# Thyroid Eye Disease as Initial Manifestation of Graves’ Disease Following Viral Vector SARS-CoV-2 Vaccine: Report of a Case and Review of the Literature

**DOI:** 10.3390/vaccines11101574

**Published:** 2023-10-10

**Authors:** Anastasia K. Armeni, Georgios Κ. Markantes, Alexandra Stathopoulou, Katerina Saltiki, Petros Zampakis, Stelios F. Assimakopoulos, Marina A. Michalaki

**Affiliations:** 1Division of Endocrinology—Department of Internal Medicine, School of Health Sciences, University of Patras, 26504 Patras, Greece; armeni@upatras.gr (A.K.A.); markantes@upatras.gr (G.K.M.); alexst71@gmail.com (A.S.); 2Endocrine Unit, Department of Clinical Therapeutics, National and Kapodistrian University, 11528 Athens, Greece; saze@otenet.gr; 3Department of Radiology, University Hospital of Patras, 26504 Patras, Greece; pzampakis@gmail.com; 4Division of Infectious Diseases—Department of Internal Medicine, School of Health Sciences, University of Patras, 26504 Patras, Greece; sassim@upatras.gr

**Keywords:** thyroid eye disease, Graves’ disease, viral vector SARS-CoV-2 vaccine, ChAdox1nCoV-19, COVID-19

## Abstract

COVID-19, a contagious disease caused by the novel coronavirus SARS-CoV-2, emerged in 2019 and quickly became a pandemic, infecting more than 700 million people worldwide. The disease incidence, morbidity and mortality rates have started to decline since the development of effective vaccines against the virus and the widespread immunization of the population. SARS-CoV-2 vaccines are associated with minor local or systemic adverse reactions, while serious adverse effects are rare. Thyroid-related disorders have been reported after vaccination for COVID-19, and Graves’ disease (GD) is the second most common amongst them. Thyroid eye disease (TED), an extrathyroidal manifestation of GD, is rarely observed post-COVID-19 vaccination. All TED cases followed mRNA-based vaccinations, but two new onset mild TED cases post-viral vector vaccine (ChAdox1nCoV-19) have also been reported. We report the case of a 63-year-old woman who presented with new onset hyperthyroidism and moderate-to-severe and active TED 10 days after she received the first dose of a viral vector vaccine against SARS-CoV-2. This is the first case of moderate-to-severe TED after such a vaccine. Our patient was initially treated with intravenous glucocorticoids, and subsequently with intravenous rituximab, due to no response. The disease was rendered inactive after rituximab, but constant diplopia persisted, and the patient was referred for rehabilitative surgery.

## 1. Introduction

COVID-19, a contagious disease caused by the novel coronavirus SARS-CoV-2, emerged in 2019 and quickly became a pandemic, infecting more than 700 million people worldwide. Despite the emergence of new SARS-CoV-2 variants, the number of new cases, as well as morbidity and mortality from COVID-19, are declining. In addition, based on the current trends, the World Health Organization estimates that by the end of 2023, the COVID-19 emergency could be ended worldwide [1]. A pivotal factor that has led to this positive perspective is the rapid response of the scientific community and pharmaceutical industry to the pandemic through the development of effective vaccines. Despite great disparities in vaccination among Europeans, at present, most EU countries have a significant immunization level. The vaccines from various platforms (mRNA, viral vectors, protein base and inactivated) have contributed to decreased incidence, severity and mortality. The SARS-CoV-2 vaccines usually induce minor local or systemic adverse effects, such as injection site pain, swelling and redness, or fatigue, fever, headaches, joint and muscle pain [2]. More severe adverse events, including anaphylaxis, myocarditis and vaccine-induced immune thrombotic thrombocytopenia, are rare [3,4,5].

Thyroid disorders post-COVID-19 vaccinations have been rarely reported [6]. Most of these cases occurred post-immunization with mRNA-based vaccinations (68.7%), followed by viral vector vaccines (15.7%) and inactivated vaccines (14.5%). Jafarzadeh et al. [6] reviewed 83 cases of thyroid abnormalities post-COVID 19 vaccinations from the literature and found that subacute thyroiditis, also known as de Quervain’s thyroiditis, was the most prevalent, accounting for 60.2% of all cases, followed by Grave’s disease (GD) (25.3% of cases). Thyroid eye disease (TED) is an extrathyroidal manifestation of GD and is a very rare complication following COVID-19 vaccination; only a few cases have been reported in the literature [7,8,9]. All of the reported TED cases followed mRNA-based vaccinations [7], but two new onset mild TED cases post-viral vector vaccine (ChAdox1nCoV-19) have also been reported [10].

GD is the most common cause of hyperthyroidism in iodine sufficient areas, with an annual incidence of 20 to 50 cases per 100,000 persons [11]. It is a humoral autoimmune disorder that causes goiter and hyperthyroidism and could be accompanied by extrathyroidal manifestations such as TED and pretibial myxedema. Forty percent of patients with GD may develop TED, but only 6% and 0.5% will have moderate-to-severe and sight-threatening disease, respectively [12,13]. The time course of TED has been described by Rundle in 1957 [14]. It has an initial inflammatory phase that lasts for 6 to 24 months, followed by a short stable phase, and then it transitions into the last fibrotic, quiescent phase [15]. Early medical intervention in the inflammatory phase (with glucocorticoids or other immunomodulatory therapies) prevents the progression of the disease, whereas such a therapy is not reasonable in the quiescent phase. The approach for each patient with TED includes the assessment of the activity and severity of the disease to plan its management. The disease activity is evaluated using several clinical tools. The most widely used is the Clinical Activity Score (CAS), adopted by the European Group on Graves’ Orbitopathy (EUGOGO) [16]. The CAS is a 7-point score combining subjective (patient symptoms) and objective (patient evaluation) evidence. The most affected eye is used to score a patient, and a CAS score ≥ 3 indicates active disease, corresponding to the TED initial phase.

Herein, we present, for the first time, a case of a new onset, moderate-to-severe TED, resistant to medical therapy, which developed within 10 days post-viral vector COVID-19 vaccine.

## 2. Case Presentation

A 63-year-old woman was referred to our tertiary outpatient endocrinology clinic in April 2022 from a private practicing endocrinologist, due to moderate-to-severe and active TED unresponsive to intravenous glucocorticoids. The woman received the first dose of the SARS-CoV-2 vaccination with a viral vector vaccine (Vaxzevria COVID-19 Vaccine ChAdOx1-S [recombinant]—previously known as COVID-19 Vaccine AstraZeneca), on 9 March 2021. She did not have a history of severe allergic reactions to any of the ingredients in the vaccine, nor did she have any other contraindication to it. Ten days later, she presented spontaneous orbital pain, redness and swelling of the eyelids, conjunctival erythema and restriction of ocular movements of her left eye, leading to constant diplopia. Thus, the TED CAS score was 4/7 and the severity of the disease was moderate-to-severe. She did not report any symptoms of hyperthyroidism before or after the onset of her eye symptoms; however, upon examination, she had tremor, tachycardia and mild goiter. She did not have a known history of Graves’ disease, other chronic diseases or recent infection, and her medical record was otherwise unremarkable. She had received vaccinations against diphtheria/tetanus/pertussis, measles/mumps/rubella and poliomyelitis in childhood, without any adverse reaction. She was a heavy smoker (90 pack-years). There was no history of autoimmune diseases or thyroid disorders in the family. At that time, her thyroid hormone levels were: TSH < 0.005 mIU/L (normal range: 0.27–4.20 mIU/L); free thyroxine (FT4): 3.5 ng/dL (normal range: 0.8–2 ng/dL), triiodothyronine (Τ3): 3 ng/mL (normal range: 0.8–2 ng/mL), Thyrotropin Receptor Antibodies (TRAbs): 4.43 U/L (normal range: <1.0 U/L). Furthermore, her serum acetylcholine receptor and muscle-specific tyrosine kinase antibodies were measured and found to be negative. Sonography of the thyroid gland revealed a non-nodular goiter, with mild hypoechogenicity and increased vascularity. Due to markedly asymmetric ophthalmopathy, she underwent an orbital MRI (1 month following the initial insult). A remarkable enlargement of the inferior rectus, and to a lesser extent, of the medial, as well as the lateral rectus of the left eye, was found. The rest of the extraocular muscles were found to be normal (Figure 1a,b). Barrett’s index (BI) was found to be 50% (strong indicator of absence of dysthyroid optic neuropathy). The patient was initially treated with methimazole (40 mg/day), propranolol and selenium (100 μg twice daily for 6 months). Intravenous glucocorticoids (IVGC) were started on May 2021 (500 mg methylprednisolone once weekly for 6 weeks and then 250 mg once weekly for another 6 weeks). Moreover, she quit smoking. Due to non-remission of the diplopia and TED activity after IVGC treatment, repeated orbital MRI was performed and the imaging findings were almost identical to those of the initial one (Figure 2a,b).

A few months later, she was referred to our clinic. At presentation, she had active disease, as the CAS score was 4/7 (spontaneous orbital pain, orbital pain upon eye movement, eyelid swelling, conjunctival redness); she also had unilateral exophthalmos of the left eye (Hertel exophthalmometer: right eye 14 mm and left eye 20 mm), ocular motility restriction and constant diplopia (Figure 3). Her thyroid function tests, under methimazole 20 mg/day and levothyroxine 125 μg/day, were: TSH: 1.77 mIU/L, T3: 0.887 ng/mL, FT4: 1.61 ng/dL and positive TRAbs: 3.87 U/L. Her serum T3, FT4, TSH and TRAbs concentrations were determined using the electrochemiluminescence immunoassay method, using Roche Diagnostic’s kits and Roche/Hitachi Cobas e-411 analyzers (GmbH, Mannheim, Germany). The liver function tests (LFT), complete blood count and renal function tests (RFT) were normal, whereas the low density lipoprotein (LDL) was 190 mg/dL. No secondary causes of hypercholesterolemia were identified from her history, clinical examination and first-line laboratory tests. The hepatitis serology and purified protein derivative (PPD) tests for tuberculosis were negative. Another orbital MRI (3rd in total) was performed to assess the severity of extraocular muscle involvement at presentation to our clinic. The imaging findings were practically unchanged when compared to the previous studies (Figure 4a,b).

The patient was administered two intravenous doses of rituximab (1 gr/dose, given 2 weeks apart), as only this type of second line treatment is available in our country. She was advised to start rosuvastatin 10 mg once daily, while she remained euthyroid under treatment with 5 mg methimazole per day; levothyroxine was discontinued. An improvement in the CAS value from 4 to 2 was noted; however, the constant diplopia remained. Thus, she was referred for surgical correction of diplopia after her TED had remained inactive for 6 months. A timeline of the key events in the patient’s course is shown in Figure 5.

## 3. Review of the Literature

To date, TED following SARS-CoV-2 vaccination has been reported in 23 patients [7,8,9,10,17,18,19,20,21,22,23], 16 of which were women (69.6%) and 7 men (30.4%), with a median age of 50 years [IQR 43–59] (Table 1). Of note, 8 out of the 23 patients (34.7%) had a personal history of autoimmunity. The mRNA vaccine was administered in 21 out of the 23 patients (91.3%), while the viral vector vaccine was administered in 2 (8.7%). Concerning the mRNA vaccine, TED manifested after the first vaccine dose in 8 out of the 21 patients (38%) and after the second dose in 12 out of the 21 patients (57%), while in one case, the timing of the TED onset was not clarified. Regarding the viral vector vaccine, both of the TED cases manifested within 2 weeks after the first dose.

The proportion of new TED cases was 56.5% (13 out of 23 patients), and the remaining 43.5% (10 out of 23 patients) concerned patients with recurrence/worsening of TED. Of the new TED cases, in 84.6% (11 out of 13 patients), TED presented following mRNA vaccine, and in 15.4%, after viral vector vaccine, while thyrotoxicosis was present in 69.2% of these cases. Concerning patients with recurrence/worsening TED, all were vaccinated with the mRNA vaccine, while thyrotoxicosis was present in 10% of them (1 out of 10). Finally, the median time interval between vaccination and TED appearance was 14 days (Table 1).

## 4. Discussion

To the best of our knowledge, this is the first description of new onset, moderate-to-severe TED along with GD, following the first dose of the viral vector ChAdOx1 COVID-19 vaccine. Our patient was a heavy smoker, middle-aged woman without a known history of GD or any other thyroid or autoimmune disease. She did not receive any medication at presentation, but severe untreated hypercholesterolemia was found. She did not have a documented previous COVID-19 infection. In the literature, only a few cases of TED post-COVID-19 vaccination have been reported; all were after mRNA-based vaccines, but two mild cases post-viral-vector have also been reported.

Orbital magnetic resonance imaging (MRI) is a very helpful tool for the initial diagnosis and follow-up of TED. It significantly helps in the assessment of disease activity and the prediction of the response to treatment [16]. Moreover, due to its quantitative evaluation, it can reliably predict dysthyroid optic neuropathy (DON) through measuring Barrett’s index (BI). This is an assessment of muscle expansion as a percentage of horizontal or vertical extraocular muscles occupied by the height or width axis [24], and it accurately predicts DON, a sight-threatening complication of TED. A BI ≥ 60% is highly sensitive and specific for DON, whereas a BI < 50% excludes optic nerve compression [24]. In our case, BI was 50%, compatible with the absence of optic nerve damage in the course of our patient’s TED.

GD is caused by the production of antibodies against thyroid-stimulating hormone (TSH) receptor (TSHR), expressed on the cell membrane of thyroid follicular cells. The antibodies stimulate the TSHR, mimicking TSH action and causing goiter and hyperthyroidism. TSHR is also expressed in the orbital adipocytes and fibroblasts [12]. TSHR antibodies activate retroocular fibroblasts, which proliferate and secrete hydrophilic glycosaminoglycans (GAG), mostly hyaluronic acid, and differentiate into mature adipocytes. TSHR is co-localized with insulin-like growth factor 1 receptor on the membranes of orbital fibroblasts, and synergistically regulate signaling cascades in TED. The accumulation of GAG in the orbital adipose tissue and extra ocular muscles leads to oedema, and in conjunction with adipogenesis, results in proptosis, diplopia and impaired venous drainage [12,25].

The initiation of the autoimmune response in GD and TED is not fully explained. Genetic and environmental triggers are implicated [26]. Thus, specific polymorphisms of several genes, such as human leukocyte antigen genes (HLA-DRβ-Arg74), the protein tyrosine phosphatase, non-receptor type 22 (PTPN22), thyroglobulin and thyrotropin, confer a susceptibility for developing GD [27]. Unfortunately, no molecular analysis was performed in our patient. Smoking, severe stress and radioactive iodine ablation of the thyroid gland are considered risk factors for TED in predisposed patients with GD [28,29,30,31]; our patient was a heavy smoker. Furthermore, hypercholesterolemia has been suggested as a risk factor for TED; therefore, the management of blood lipid levels should be included in the management of TED [32]. The culpability of the vaccination to the development of TED could not be unequivocally proven in our patient. However, the temporal correlation may be indicative of such a relation, similar to all of the other cases in the literature, where the causality was suspected due to the appearance of TED within a few days following vaccination (10 days in our case, a median of 14 days in the literature). In our patient, as well as in all of the reported cases of TED after COVID-19 vaccination, no other triggering event (e.g., radioactive iodine treatment, major illness/surgery, pregnancy) other than the vaccine itself was identified. In most of the reported cases, the TED was characterized as moderate-to-severe, like in our case. Furthermore, almost half of the reported patients had pre-existing GD (Table 1). It should be noted that we cannot exclude the possibility of pre-existent undiagnosed GD in our patient, as the disease is 5–10 times more prevalent in women [33]. Viral or bacterial infections could precede autoimmune diseases, and hepatitis C virus, enterovirus and reovirus have been postulated to trigger GD [33]. No such history or laboratory evidence was found in our case.

On the other hand, vaccines contain inactivated viruses or share viral antigens, so they may cause autoimmune reactions. Several SARS-CoV-2 proteins, such as the spike protein, the nucleoprotein and the membrane protein, have an homology with the enzyme thyroidal peroxidase (TPO); thus, they could trigger GD and/or TED via molecular mimicry in predisposed subjects [34]. Another potential pathogenic mechanism could be the direct injury of follicular cells by SARS-CoV-2, leading to the release of antigens and the initiation of the autoimmune response. The thyroid epithelial cells express the receptor used by SARS-CoV-2 to enter cells, namely the angiotensin-converting enzyme 2 (ACE2), as well as a transmembrane protease, the serine 2 (TMPRSS2), thus facilitating virus infectivity.

Lastly, the vaccine’s adjuvants can lead to an overstimulation of the immune system, resulting in the well-known autoimmune/inflammatory syndrome (ASIA), described in 2011 [35]. Most types of vaccines also contain adjuvants, which may enhance the immune response to the infectious antigen and, consequently, could provoke autoimmunity. mRNA-based vaccines lack adjuvants [36]; thus, the basis for this hypothesized immune system overstimulation is not known. Some authors suggest that mRNA has intrinsic immunostimulatory properties, while another speculation is that lipid nanoparticles may act as adjuvants, triggering the autoimmunity processes [37].

TED treatment depends on the disease activity and severity. According to the latest joint guideline by the American Thyroid Association and the European Thyroid Association, the restoration of euthyroidism, smoking cessation and local measures are appropriate for all patients. Selenium should be considered in patients with mild, active TED, while intravenous glucocorticoids are the preferred treatment for active moderate-to-severe TED [13]. In patients who are unresponsive to glucocorticoids, rituximab (a monoclonal antibody against CD20) and tocilizumab (an interleukin-6 receptor blocker) should be considered to achieve TED inactivation. Another monoclonal antibody that targets the insulin-like growth factor 1 receptor, teprotumumab, is the preferred therapy in TED characterized by significant proptosis and/or diplopia [13]. Of these second-line treatments, rituximab is currently the only available treatment in our country. By binding to CD20, rituximab downregulates B cell function and antibody production, and the available evidence suggests that it is efficacious in inactivating TED (even in glucocorticoid-resistant patients) and in preventing TED relapses for over 1 year [13]. Based on the above, we decided to administer rituximab to our patient who had moderate-to-severe TED unresponsive to IVGC; following this intervention, TED inactivation was achieved and maintained.

The main limitation of this work is that the link between vaccination and TED development cannot be unequivocally established. Nevertheless, this case, along with the others reported in the literature, suggests that COVID-19 vaccines might induce the appearance or worsening of GD and/or TED. Clinicians should be aware of this possibility in order not to overlook symptoms of these conditions and to offer their patients the appropriate treatments.

In conclusion, we described a very rare case of moderate-to-severe TED as the initial manifestation of GD following the first shot of the ChAdOx1 COVID-19 vaccine, unfortunately with an unfavorable outcome. Although the causal relationship between the vaccination and TED cannot be unequivocally proven, clinicians should be aware of such potential autoimmune side effects of the COVID vaccination.

## Figures and Tables

**Figure 1 vaccines-11-01574-f001:**
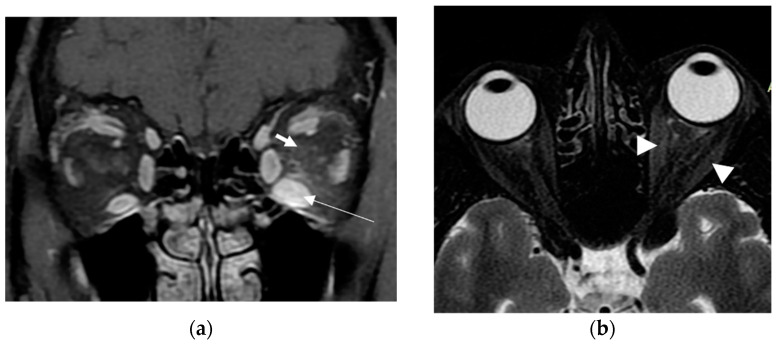
First MRI. (**a**) Coronal T1W post-GD fat-suppressed MRI at a point halfway between the posterior globe and the orbital apex, depicts marked enlargement of the inferior rectus muscle (white long thin arrow) as well as subtle high signal within the retrobulbar fat (white short thick arrow); (**b**) Axial T2W fat-suppressed MRI at the level of optic nerve, shows minor enlargement of medial and lateral rectus muscles (white arrowheads).

**Figure 2 vaccines-11-01574-f002:**
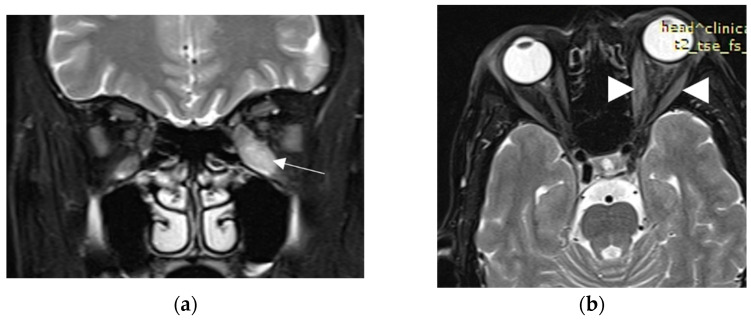
Second MRI. (**a**) Coronal T2W fat-suppressed MRI at a point halfway between the posterior globe and the orbital apex, reveals persistent enlargement and high signal of inferior rectus muscle (white arrow); (**b**) Axial T2W fat-suppressed MRI at the level of optic nerve, depicts persistent minor enlargement of medial and lateral rectus muscles (white arrowheads).

**Figure 3 vaccines-11-01574-f003:**
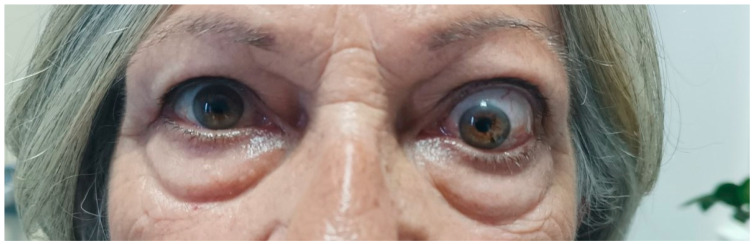
Photograph of the patient at presentation to our clinic. Left-sided exophthalmos, eyelid swelling, conjunctival redness and restriction of ocular motility are noted.

**Figure 4 vaccines-11-01574-f004:**
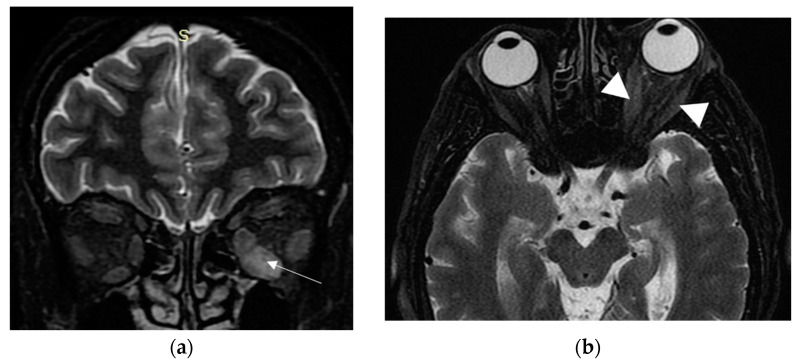
Third MRI. (**a**) Coronal T2W fat-suppressed MRI at a point halfway between the posterior globe and the orbital apex, reveals persistent enlargement and high signal of inferior rectus muscle (white arrow); (**b**) Axial T2W fat-suppressed MRI at the level of optic nerve, depicts persistent minor enlargement of medial and lateral rectus muscles (white arrowheads).

**Figure 5 vaccines-11-01574-f005:**
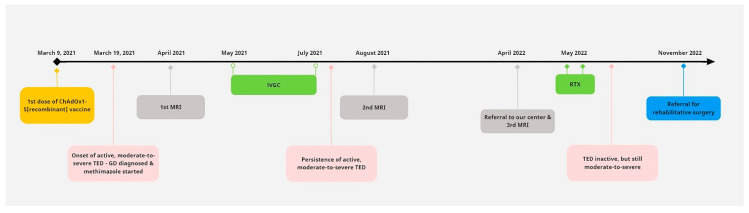
Timeline of the key events in the patient’s course. TED: thyroid eye disease, GD: Graves’ disease, IVGC: intravenous glucocorticoids (500 mg methylprednisolone once weekly for 6 weeks and then 250 mg once weekly for another 6 weeks), RTX: intravenous rituximab (two doses of 1 gr, given 2 weeks apart).

**Table 1 vaccines-11-01574-t001:** Clinical and hormonal data as well as activity and severity of TED in patients with TED post-SARS-CoV-2 vaccination.

Sex	Age	Family History of Autoimmunity	Personal History of Autoimmunity	Name of Vaccine	Type of Vaccine	Dose	Time of TED Onset after Vaccination (Days)	History of TED	Thyrotoxicosis	CAS	Severity of TED	Reference
F	34	ND	ND	Pfizer	mRNA	1st	10	New	Yes	ND	Mild	[17]
F	50	ND	ND	Pfizer	mRNA	2nd	3	New	No	5/7	Moderate-to-severe	[8]
F	51	ND	No	Pfizer	mRNA	2nd	4	New	Yes	3/7	Mild GO	[18]
F	71	ND	No	Pfizer	mRNA	2nd	>70	New	Yes	ND	Moderate-to-severe / 10 weeks after treatment of Graves’ disease	[19]
M	43	ND	ND	Pfizer	mRNA	ND	14	TED recurrence	No	ND	Sight threatening	[9]
F	58	ND	ND	Pfizer	mRNA	2nd	3	Worsening of 3-year TED	No	6/10	Moderate-to-severe	[9]
F	53	ND	ND	Pfizer	mRNA	1st	1	New	No	ND	Moderate-to-severe	[20]
F	66	ND	ND	Moderna	mRNA	2nd	21	TED recurrence after 15 years	No	6/10	Moderate-to-severe	[20]
F	37	ND	ND	ND	mRNA	2nd	21	New	Yes	ND	Mild-to-moderate	[21]
F	45	ND	ND	Moderna	mRNA	1st	21	TED recurrence after 5 years	ND	ND	Mild-to-moderate	[20]
M	59	ND	ND	ND	mRNA	1st	21	New	Yes	ND	Mild-to-moderate	[21]
F	34	ND	ND	ND	mRNA	1st	26	New	Yes	ND	Mild-to-moderate	[21]
F	70	ND	Graves’ disease (total thyroidectomy)	ND	mRNA	2nd	6	TED recurrence after 18 weeks	No	4/7	Moderate-to-severe	[7]
M	43	ND	Type-1 diabetes mellitus, psoriasis, Graves’ disease	Moderna	mRNA	1st	1	TED recurrence after 45 weeks	No	7/7	Sight threatening	[7]
M	73	ND	Graves’ disease	Pfizer	mRNA	1st	21	New	No	3/7	Mild	[7]
F	45	ND	Graves’ disease (total thyroidectomy)	Moderna	mRNA	2nd	7	TED recurrence	No	4/7	Moderate-to-severe	[7]
M	48	ND	Graves’ disease (total thyroidectomy)	Moderna	mRNA	2nd	30	TED recurrence after 11 months	Yes	5/7	Sight threatening	[7]
F	39	Thyroid disease	ND	Pfizer	mRNA	1st	7	New	No	2/7	Mild	[7]
M	20	No	No	COVISHIELD	viral vector	1st	14	New	Yes	2/7	Mild	[10]
F	46	Hypothyroidism	No	COVISHIELD	viral vector	1st	10	New	Yes	2/7	Mild	[10]
M	50	ND	Psoriasis, vitiligo, atrophic gastritis, Graves’ disease	Pfizer	mRNA	2nd	21	Worsening of 9-months TED	No	7/10	Moderate-to-severe	[22]
F	71	ND	Hypothyroidism	Moderna	mRNA	2nd	3	New	Yes	4/7	Sight threatening	[22]
F	51	ND	Graves’ disease	Moderna	mRNA	2nd	15	TED recurrence after 30 weeks	No	9/10	Severe	[23]

ND: not documented.

## Data Availability

The data are available upon request.

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
