# Peer review of "Thyroid Eye Disease as Initial Manifestation of Graves’ Disease Following Viral Vector SARS-CoV-2 Vaccine: Report of a Case and Review of the Literature"

_vaccines, 2023, doi:10.3390/vaccines11101574_

Round 1

Reviewer 1 Report

The study's objective is compelling, but its elaboration needs refinement, particularly in how Table 1 is presented. It might be better placed in the discussion section. At present, the format resembles a mini-review. The author should focus on highlighting their findings, using Table 1 for support. While the observational analysis in figures 1 to 3 is satisfactory, Table 1 lacks significance. A substantial revision is recommended.

Author Response

REVIEWER 1

“The study's objective is compelling, but its elaboration needs refinement, particularly in how Table 1 is presented. It might be better placed in the discussion section. At present, the format resembles a mini-review. The author should focus on highlighting their findings, using Table 1 for support. While the observational analysis in figures 1 to 3 is satisfactory, Table 1 lacks significance. A substantial revision is recommended”.

Thank you very much for your comments. Our aims, as stated in the manuscript title, were to present our case and to provide a review of the relevant literature. That is why we included a separate paragraph entitled “Review of the literature” and summarized all reported cases of TED following vaccination against COVID-19 in Table 1. Certainly, data from table 1 is used to support our findings and now a new paragraph has been added to the discussion section (lines 261-266). We have also added more information regarding the characteristics of our patient (lines 100-101 and 104-109), and a timeline to better illustrate the evolution of the patient’s condition (Figure 5).

Reviewer 2 Report

The study expands the number of cases that link the development of grave’s disease and thyroid eye disease with the application of a vaccine based on viral vectors. The study provides a case in which the effects persisted even after the commonly applied treatment. The study is of interest to vaccinology so I consider that it meets the criteria to be published.

Author Response

REVIEWER 2

“The study expands the number of cases that link the development of grave’s disease and thyroid eye disease with the application of a vaccine based on viral vectors. The study provides a case in which the effects persisted even after the commonly applied treatment. The study is of interest to vaccinology so I consider that it meets the criteria to be published”.

Thank you very much for your comments.

Reviewer 3 Report

The case report is clear, concise and well-written, aiming at a rare complication of COVID vaccine. I have only three comments:

1.Because thyroid eye disease was the initial manifestation of Graves' disease, it is important to report any symptoms of hyperthyroidism before the onset of thyroid eye disease.

2.Since the patient underwent MRI for three times and the findings were identical, I suggest to add 1st, 2nd, and 3rd in figure legends in Figure 1, 2 and 4, respectively.

3.Cholesterol level tend to be low in patients with hyperthyroidism, did the patient have any secondary cause of hypercholesterolemia ?

Author Response

REVIEWER 3

“The case report is clear, concise and well-written, aiming at a rare complication of COVID vaccine. I have only three comments:

1.Because thyroid eye disease was the initial manifestation of Graves' disease, it is important to report any symptoms of hyperthyroidism before the onset of thyroid eye disease.

2.Since the patient underwent MRI for three times and the findings were identical, I suggest to add 1st, 2nd, and 3rd in figure legends in Figure 1, 2 and 4, respectively.

3.Cholesterol level tend to be low in patients with hyperthyroidism, did the patient have any secondary cause of hypercholesterolemia?”

Thank you very much for your comments.

  1. The patient did not report symptoms of hyperthyroidism before or after the onset of eye symptoms. We have added this information in section 2 (lines 104-106).
  2. Done, thank you.
  3. No secondary cause of hypercholesterolemia was apparent after history taking, clinical examination and basic laboratory tests. A comment has been added in section 2 (lines 160-161).

Reviewer 4 Report

This case report discusses a 63-year-old woman who developed hyperthyroidism and moderate-to-severe thyroid eye disease (TED) following her first dose of a viral vector vaccine against SARS-CoV-2, the virus responsible for COVID-19. While cases of thyroid-related disorders, including Graves' disease (GD), have been reported after COVID-19 vaccination, this is the first reported instance of moderate-to-severe TED occurring after such a vaccine. The patient initially received treatment with intravenous glucocorticoids and later intravenous rituximab, which rendered the disease inactive but did not alleviate constant diplopia (double vision). Consequently, the patient was referred for rehabilitative surgery. I suggest adding medical record of the patient, any pre-existing conditions, and any other vaccination status. 

Information about the patient's overall health and any possible contraindications or known vaccine side effects would be valuable to add in the case presentation. This can help in establishing a link between the vaccine and the development of TED.

The report provides information at multiple points in the patient's journey but lacks a comprehensive overview of her condition over time. Including a timeline with key events and treatment responses can help understanding the disease's progression and treatment efficacy.

The report does not provide information regarding the potential mechanisms behind the development of TED following vaccination. A discussion of the immunological processes or pathways involved could provide insights for researchers and clinicians.

Discussing the use of rituximab as a second-line treatment option in more detail, along with its rationale, could be informative for clinicians considering similar cases. 

Write limitations of the case report and future recommendation for doctor if they receive such cases must be included.

Author Response

REVIEWER 4

“This case report discusses a 63-year-old woman who developed hyperthyroidism and moderate-to-severe thyroid eye disease (TED) following her first dose of a viral vector vaccine against SARS-CoV-2, the virus responsible for COVID-19. While cases of thyroid-related disorders, including Graves' disease (GD), have been reported after COVID-19 vaccination, this is the first reported instance of moderate-to-severe TED occurring after such a vaccine. The patient initially received treatment with intravenous glucocorticoids and later intravenous rituximab, which rendered the disease inactive but did not alleviate constant diplopia (double vision). Consequently, the patient was referred for rehabilitative surgery”.

“I suggest adding medical record of the patient, any pre-existing conditions, and any other vaccination status”.

The relevant information has been added in section 2 (lines 107-109), thank you.

“Information about the patient's overall health and any possible contraindications or known vaccine side effects would be valuable to add in the case presentation. This can help in establishing a link between the vaccine and the development of TED”.

The patient did not have any contraindications to the received vaccine or known vaccine side-effects. The relevant information has been added in section 2 (lines 100-101 and 108-109), thank you.

“The report provides information at multiple points in the patient's journey but lacks a comprehensive overview of her condition over time. Including a timeline with key events and treatment responses can help understanding the disease's progression and treatment efficacy”.

Thank you for your comment. A timeline has been added, see Figure 5.

“The report does not provide information regarding the potential mechanisms behind the development of TED following vaccination. A discussion of the immunological processes or pathways involved could provide insights for researchers and clinicians”.

Thank you for your comment. At present, the molecular mechanisms behind vaccination-related TED development have not been fully clarified. In the discussion section of our manuscript, we present the potential mechanisms proposed so far, namely the spike protein and the vaccine adjuvants hypotheses (lines 271-287). Furthermore, in our case as in all the reported cases of TED after COVID-19 vaccination, the only indications of a causal relationship are the temporal correlation (manifestation of TED within a few days of vaccination) and the lack of other triggering events (see the addition we have made, lines 261-266). 

“Discussing the use of rituximab as a second-line treatment option in more detail, along with its rationale, could be informative for clinicians considering similar cases”.

Thank you for your suggestion. We have added a paragraph in the Discussion section of the manuscript (lines 288-303).

“Write limitations of the case report and future recommendation for doctor if they receive such cases must be included”.

Done (lines 304-308), thank you.